# Giant Cloud Condensation Nuclei enhanced Ice Sublimation Process: A potential mechanism in mixed phase clouds

Denghui Ji<sup>1</sup>, Christoph Ritter<sup>2</sup>, Xiaoyu Sun<sup>1</sup>, Manuel Moser<sup>3,4</sup>, Christiane Voigt<sup>3,4</sup>, Mathias Palm<sup>1</sup>, and Justus Notholt<sup>1</sup>

**Correspondence:** Xiaoyu Sun (xiaoyu\_sun@iup.physik.uni-bremen.de)

Abstract. The Wegener-Bergeron-Findeisen (WBF) process describes the growth of ice crystals at the expense of supercooled liquid droplets in mixed-phase clouds, driven by phase transitions at temperatures below 0°C. In this study, we introduce a potential mechanism involving the transfer of water vapor from ice to cloud droplets formed on Giant Cloud Condensation Nuclei (GCCN). This process occurs under specific atmospheric conditions influenced by temperature and CCN size, particularly for CCN with diameters exceeding 1 µm. We term this mechanism the Giant Cloud Condensation Nuclei-Enhanced Ice Sublimation Process (GCCN-ISP). We first conduct a theoretical analysis to develop a physical model for determining these specific atmospheric conditions, followed by validation through observations. Model simulations informed by observational data from aircraft indicate that when CCNs are sufficiently large and cold, the water vapor partial pressure over droplets formed on these CCNs can be lower than that over ice. Consequently, water vapor can transfer from ice to supercooled droplets, causing the droplets to grow. Eventually, the water vapor pressures of both reach equilibrium, resulting in their coexistence.

# 1 Introduction

Mixed-phase clouds, characterized by the coexistence of both ice crystals and supercooled liquid droplets, represent a complex and interesting phenomenon in atmospheric science (Korolev et al., 2017). Mixed phase clouds exhibit unique radiative properties due to the presence of both ice and liquid phases, which affect the balance of incoming and outgoing radiation in the atmosphere (Matus and L'Ecuyer, 2017; D'Alessandro et al., 2019; Yan et al., 2020). Moreover, mixed-phase clouds contribute to feedback mechanisms that amplify climate change in the Arctic. Changes in cloud properties can alter surface albedo (Jonsell et al., 2003), affecting the absorption and reflection of solar radiation (Hogan et al., 2003; Ehrlich et al., 2008; Matus and L'Ecuyer, 2017). This, in turn, influences temperature patterns (Pithan et al., 2014), and sea ice extent (Shupe et al., 2013; Villanueva et al., 2022) in the Arctic region. Therefore, understanding microphysical processes in mixed-phase clouds is crucial for comprehending climate processes and feedback mechanisms.

Mixed-phase clouds play an important role in climate change, yet studying them remains challenging due to observational limitations (Korolev et al., 2017) and the complexities of accurately modeling their microphysical and dynamical properties

<sup>&</sup>lt;sup>1</sup>Institute of Environmental Physics, University of Bremen, Otto-Hahn-Allee 1, 28359 Bremen, Germany

<sup>&</sup>lt;sup>2</sup>Alfred Wegener Institute, Helmholtz Centre for Polar and Marine Research, Telegrafenberg A43, 14473 Potsdam, Germany

<sup>&</sup>lt;sup>3</sup>Institut für Physik der Atmosphäre, Johannes Gutenberg-Universität, Mainz, Germany

<sup>&</sup>lt;sup>4</sup>Institut für Physik der Atmosphäre, Deutsches Zentrum für Luft- und Raumfahrt, Weßling, Germany

(Bodas-Salcedo et al., 2014; McCoy et al., 2015). Observations from satellite studies (Gryspeerdt et al., 2014; Mayer et al., 2024) and in-flight campaigns (Kirschler et al., 2023; Moser et al., 2023; Crosbie et al., 2024) confirm the ubiquitous presence of mixed-phase clouds in the lower and mid-troposphere across mid- and high-latitudes, providing valuable insights into their behavior and interactions (Chellappan et al., 2024). Thus, unraveling the underlying physical mechanisms governing mixed-phase clouds is essential for improving climate models and forecasting capabilities, ultimately advancing our understanding of Arctic and global climate change.

Fundamentally, the Wegener-Bergeron-Findeisen (WBF) process, a cornerstone of cloud physics, describes a growing of the ice and evaporation of supercooled liquid water within a specific temperature range between -38 and 0 °C (Wegener, 1911; Bergeron, 1928, 1935; Findeisen, 1938). Based on WBF process, theoretically, there are four different scenarios of phase transformation in mixed-phase clouds (Korolev et al., 2017): 1) both ice particles and droplets evaporate, whereas the mass of the vapor increases. In terms of the vapor pressure, this corresponds to the condition when  $e < e_i < e_l$ , where the ambient water vapor pressure is donated as e. Here,  $e_i$  and  $e_i$  are the saturation vapor pressure over liquid and ice, respectively; 2) ice particles grow, droplets evaporate, and the water vapor mixing ratio increases; 3) ice particles grow, droplets evaporate, and the water vapor mass decreases; 4) both ice particles and liquid droplets grow, and the water vapor mass decreases ( $e_i < e_l < e$ ). Those scenarios of phase transformation in mixed-phase clouds assume that solute effects are neglected (Fan et al., 2011; Korolev et al., 2017), which is primarily because the size of typical cloud condensation nuclei (CCN) is too small to significantly influence the surface saturated saturation water vapor pressure over a liquid cloud droplet. Chen et al. (2023) demonstrate that for a CCN with a radius of 1 μm, the solute effect becomes negligible when it forms a cloud droplet with a radius of 10 μm. In mixed-phase clouds, droplets size can vary significantly, with droplets exceeding 10 um in radius, but smaller droplets have also been observed (Moser et al., 2023). Atmospheric aerosol particles are typically smaller than 1 µm (Boyer et al., 2023), yet certain CCNs, such as sea salt aerosols from the open ocean or blowing snow in the Arctic, can reach diameters as large as 10 µm (Huang and Jaeglé, 2016; Yang et al., 2018; Frey et al., 2020; Gonzalez et al., 2023). This variability in particle size highlights the diverse microphysical processes at play in mixed-phase clouds, influencing their role in cloud-aerosol interactions and climate dynamics. When the size of the CCN is large, leading to a significant enhancement of the solute effect and decreasing the vapor pressure of the liquid particles, the phase transfer of moisture in mixed-phase clouds may take a different direction, such as the ice-to-liquid water transfer that we propose. This study aims to explore this potential new process by theoretically examining the saturated saturation water vapor pressure of supercooled water droplets while considering strong solute effects. We refer to this process as the Giant Cloud Condensation Nuclei enhanced Ice Sublimation (GCCN-ISP).

To achieve this goal, we present a detailed theoretical framework in Section 2. Subsequently, Section 3 validates our theoretical findings through aircraft in-situ measurements. Finally, Section 4 presents the results of our study, offering insights into the behavior of mixed-phase clouds and the implications for climate modeling and forecasting capabilities.

## 2 Theoretical Framework

55

75

## 2.1 Saturated Saturation Water Vapor Pressure for A Single Supercooled Liquid Droplet

Before considering large CCN in mixed phase droplets, here, we start with the saturation vapor pressure over a plane pure water (liquid) surface. The August–Roche–Magnus formula provides a very good approximation in the saturation vapor pressure over water (Alduchov and Eskridge, 1996):

$$e_l(T) = 6.1094 exp(\frac{17.625T}{T + 243.04}) \tag{1}$$

where  $e_l$  is in hPa, and T is in degree Celsius. The saturation vapor pressure over ice is given as follows (Huang, 2018):

$$e_i(T) = 6.1121 exp(\frac{22.578T}{T + 273.86}) \tag{2}$$

Figure 1a present the saturation vapor pressure over liquid water (red line) and ice (black line) as well as their difference with temperature (see the red line in Fig.1b). If the solute effect is not considered, i.e. when the ambient vapor pressure falls between the saturation vapor pressure over water and the lower saturation vapor pressure over ice, the <u>sub-saturated sub-saturation</u> environment for liquid water but a supersaturated environment for ice will result in rapid evaporation of liquid water and rapid ice crystal growth through vapor deposition. This is one possible way for water deposition in WBF Process. However, if the solute effect of the CCN is taken into account, the <u>saturated saturation</u> vapor pressure over a droplet changes.

When it comes to droplets, two effects should be considered: the Kelvin effect (Thomson, 1871) and Raoult's law (Raoult, 1887). The Kelvin effect describes the change in vapor pressure due to a curved liquid–vapor interface, such as the surface of a droplet. The vapor pressure at a convex curved surface is higher than that at a flat surface. Raoult's law relates the saturation vapor pressure to the solute. The two effects are combined, and this is the main point mentioned in the Köhler theory (Köhler, 1936), which describes the process in which water vapor condenses and forms liquid droplets. In addition, the composition of CCN is often a mixture of several components, then we can introduce  $\kappa$ -Köhler theory (Petters and Kreidenweis, 2007) to calculate this situation:

$$e_{solu}(D_p) = e_l \frac{D_p^3 - D_d^3}{D_p^3 - (1 - \kappa) D_d^3} exp(\frac{A}{D_p} - \frac{B}{D_p^3}),$$

$$\kappa = \sum_i \epsilon_i \kappa_i;$$

$$A = \frac{4M_w \sigma_w}{RT \rho_w};$$

$$B = \frac{6n_s M_w}{\pi \rho_w}.$$
(3)

where  $e_{solu}$  is the droplet solution water vapor pressure;  $e_l$  is the corresponding saturation vapor pressure over a flat surface;  $\sigma_w$  is the droplet surface tension;  $\rho_w$  is the density of pure water;  $n_s$  is the moles of solute;  $M_w$  is the molecular weight of water;  $D_p$  is the cloud droplet diameter;  $D_d$  is the diameter of the dry aerosol particle;  $\kappa$  is the hygroscopicity parameter, varying with aerosol composition. Usually,  $\kappa$  is about 1.4 for the most hygroscopic aerosols in the atmosphere, e.g. sodium chloride (Petters and Kreidenweis, 2007), and lower values of  $\kappa$  indicate less-hygroscopic, e.g. 0.05 for soot (Grimonprez et al., 2021).  $\epsilon_i$  is the proportion of each aerosol component.

Figure 1a illustrate the saturated saturation vapor pressure over droplets (20 µm in diameter) with varying CCN diameters (0.1, 1, 5, and 10 um). The difference between those droplets cases and the saturated saturation water vapor over the ice surface is shown in Fig.1b. Assuming the CCN in each droplet is a perfect sphere of NaCl, the saturated saturation water vapor pressure corresponding to these CCNs with a wet diameter of 20  $\mu$ m is then calculated. When the CCNs are small (0.1 – 1 µm), the saturated saturation vapor pressure over droplets closely resembles that of pure water, indicating a minimal influence of solute due to the dilution effect caused by a large amount of liquid water. However, as the amount of salt increases, such as with 10 µm CCNs, the saturated saturation vapor pressure of droplets becomes equal to that of the ice surface at temperatures around -17 °C (compare the yellow cross mark in the Fig.1b). This suggests that at this temperature, liquid droplets with 20 μm diameter and containing the amount of salt as the 10 μm diameter can coexist with ice as long as the ambient water vapor does not change. In other words, when a particle of sodium chloride, 10 µm in diameter, is suddenly placed in a ice cloud at a temperature of -17 °C, the CCN particle absorbs water from the ice until it grows to a 20-µm droplet, thus aligning itself with the saturated saturation water vapor pressure of the ice. At this point, the ice water no longer decreases and the liquid droplets are stabilized. Therefore, a supercooled liquid droplet with a 20 µm diameter and a 10 µm diameter salt content is termed a balanced diameter for sustaining supercooled liquid droplets with ice, thereby prolonging the lifetime of mixed-phase clouds. This process in this study is so called "GCCN-ISP". The balanced diameter, i.e. the yellow cross mark in the Fig.1b, in this study means that at a specific temperature and a specific CCN size, there exists a diameter of supercooled liquid droplets at which the saturation water vapor pressure of this supercooled droplet is the same as that of the ice. Therefore, creating a lookup table of balanced diameter, where supercooled droplets of varying CCN sizes at different temperatures coexist with ice, would be helpful in understanding the GCCN-ISP.

Combined Combining both saturation vapor pressure with respect to pure water (Eq.1) and Köhler theory (Eq.3) for sea salt with  $\kappa = 1.4$ , a modified function is given in this study:

$$e_{solu}(D_p) = e_l exp(\frac{A}{D_p} - \frac{B}{D_p^3}) = 6.1094 exp(\frac{17.625T}{T + 243.04}) exp(\frac{A}{D_p} - \frac{B}{D_p^3})$$
(4)

Let  $e_{solu}(D_p) = e_i$ , we get the equation of balanced diameter between droplets and ice surface:

100

105 
$$e_{solu}(D_p) = 6.1094 exp(\frac{17.625T}{T + 243.04}) exp(\frac{A}{D_p} - \frac{B}{D_p^3}) = 6.1121 exp(\frac{22.578T}{T + 273.86})$$
 (5)

In general, the coarse aerosol ( $> 1 \, \mu m$ ) is more likely to be sea salt (NaCl (Kirpes et al., 2018)) than sulfate, so in the following part of the work we will focus on sea salt in the analysis. According to this relationship, for a specific diameter of CCN particle under a certain temperature, the balanced diameter of a droplet with NaCl as its CCN can be calculated. Figure 2a presents the balanced diameter between droplets and ice surface under different ambient temperature with different dry aerosol size ( $2 \, \mu m - 20 \, \mu m$ ). This means that in the case of coexistence of ice and supercooled droplets, there are special droplets, of which the saturated saturation water vapor pressure equals the saturated saturation water vapor pressure on the ice surface. This allows liquid water and ice crystals to coexist stably. Therefore, Fig.2a can be treated as a lookup table for a specific CCN diameter and a specific cloud temperature. The composition in Fig.2a is NaCl. To present the difference of balanced diameter with respect to different aerosol composition, Fig.2b shows the ratio of droplet diameter in equilibrium state to dry aerosol diameter for

different aerosol compositions (sodium chloride, ammonium sulphate, and ammonium chloride). For any mixed-phase cloud, once the cloud's internal temperature is determined, as indicated by the orange arrow in Fig.2a (e.g., −17°C), and the size of the supercooled droplets is known (the blue arrow, 20 μm), a supercooled droplet of this size would require a CCN with a diameter of 10 μm to coexist with ice at that temperature. In turn, if the balanced diameter and the diameter of the contained solute can be given, the composition of the solute contained in the droplet can be deduced backwards.

# 120 2.2 Calculating the droplets size bin concentration with Consideration of Balanced Diameter

130

140

For a single supercooled droplet, as we mentioned before, its balanced diameter with respect to the ice surface theoretically exists. A mixed phase cloud has droplet particles of different sizes and differences in the solute (or different size of cloud droplet condensate) contained in each droplet particle. Therefore, the effect of a single particle with a balanced diameter within a swarm of particles needs to be theoretically demonstrated.

During the WBF process, liquid droplets in an isolated environment (no new water supply by turbulent mixing) will evaporate liquid water to ensure ice particles increases. The size of liquid droplets decreases changes at a rate as follows:

$$\begin{cases}
\frac{dr}{dt} = \frac{D\rho_v(\infty)}{r\rho_{liq}e(\infty)} [e(\infty) - e(r)] = \frac{G_t S}{r} \\
S = \frac{e(\infty) - e(r)}{e(\infty)} \\
G_l = \frac{D\rho_v(\infty)}{\rho_{liq}}
\end{cases} \tag{6}$$

Where r means radius of cloud droplet; D is diffusion coefficient of water vapor. From this equation, the radius of cloud droplet decreases changes at a rate inversely proportional to the radius:  $dr/dt \propto 1/r$ . As mentioned earlier, in this equation, there is no new water supplied by turbulent mixing, meaning that updrafts are not considered. In atmospheric clouds, the evolution of cloud droplet size is governed by two key factors: (1) the ambient water vapor pressure,  $e(\infty)$ , which determines the vapor availability for condensation or deposition, and (2) the equilibrium saturation vapor pressure over the droplet surface, e(r), which is influenced by droplet size, curvature, and composition (via the solute effect).

Updrafts affect cloud microphysics by cooling air parcels and enhancing supersaturation, thereby suppressing the WBF process and delaying ice gorwth (Khain et al., 2021; Guti'errez and Furtado, 2023; Abade and Albuquerque, 2024). In contrast, the solute effect reduces the surface saturation vapor pressure, directly influencing whether supercooled droplets can persist. This study intentionally excludes vertical motion to isolate the microphysical contribution of the solute effect to droplet–ice vapor competition.

Therefore, during WBF process, the number concentration of each cloud droplet size bin will increase due to the evaporation of droplets in larger bin. On the other hand, the number concentration of droplets in each bin will decrease due to the evaporation of droplets in the bin itself. Only in the last bin (the largest bin), the number concentration will decrease because no larger droplets will evaporate water vapor and becomes a member of this largest bin. Based on that, we assume that for a specific observation, the supercooled water droplets in the mixed phase cloud are divided into several bins, called bin 1, 2, 3,..., I, respectively. The decrease of the number concentration of last bin is  $\Delta N$  during the period,  $\Delta T$ . The relationship can be

145 written as follows:

$$\Delta N = \frac{\Delta R}{R_{I,max} - R_{I,min}} \cdot N_I = \frac{\Delta R}{\Delta L_I} \cdot N_I \tag{7}$$

Where  $\Delta R$  is the radius decrease during this time period;  $R_{I,max}$  is the maximum particle radius in the bin I and  $R_{I,min}$  is the minimum radius in the bin I.  $N_I$  is the number concentration of the cloud droplet bin I;  $\Delta L_I$  means the width of the cloud droplet bin I. Then, the radius of i-th bin, will decrease as following equation:

$$\begin{vmatrix}
\frac{\Delta R_I}{\Delta T} = \frac{Constant}{R_I} \\
\frac{\Delta R_i}{\Delta T} = \frac{Constant}{R_i}
\end{vmatrix} \Rightarrow \frac{\Delta R_i}{\Delta T} = \frac{\Delta R_I}{\Delta T} \cdot \frac{R_I}{R_i} \Rightarrow \Delta R_i = \Delta R_I \cdot \frac{R_I}{R_i}$$
(8)

Then, the number concentration of i-th bin will decrease as following equation:

$$\frac{\Delta R_i = \Delta R_I \cdot \frac{R_I}{R_i}}{\Delta N_i = \frac{\Delta R_i}{\Delta L_i} \cdot N_i} \Rightarrow \Delta N_i = \frac{\Delta R_I}{\Delta L_i} \cdot \frac{R_I}{R_i} \cdot N_i \tag{9}$$

Finally, we get the number concentration of i-th bin:

$$\begin{cases} \Delta N_i = -\frac{\Delta R_I}{\Delta L_i} \cdot \frac{R_I}{R_i} \cdot N_i & +\frac{\Delta R_I}{\Delta L_{i+1}} \cdot \frac{R_I}{R_{i+1}} \cdot N_{i+1}, i = 1, 2, 3, ..., I - 1; \\ \text{decrease due to evaporation} & \text{increase due to evaporation in next larger bin} \end{cases}$$

$$\Delta N_I = -\frac{\Delta R_I}{\Delta L_I} \cdot \frac{R_I}{R_I} \cdot N_I.$$

$$(10)$$

The above set of equations applies to mixed phase clouds with small particle aerosols (less than 0.1  $\mu$ m) acting as condensation nuclei, but if we consider even coarser aerosol particles, such as those above 1  $\mu$ m, then the supercooled droplets formed by the large particles will be at the balanced diameter, which can hold their liquid water during WBF process. This occurs because the saturated saturation water vapor pressure at the surface of the solution droplet equals that at the surface of the ice ( $e_{solu} = e_i$ ), indicating equilibrium state, and thus halting the transfer of water from the droplet to the ice crystal.

Based on that, during the WBF process, supercooled water droplets formed by small aerosol particles continuously lose water, providing the moisture needed for the growth of ice crystals, as dictated by the vapor pressure relationship  $(e_i < e_l)$ . However, for supercooled water droplets formed by larger aerosol particles, as they evaporate and gradually become concentrated solute solutions, they can reach an equilibrium state where the surface vapor pressure equals the saturation vapor pressure over ice  $(e_{solu} = e_i)$ . In this case, the solute effect allows these droplets to coexist with ice crystals. Therefore, only part of the whole droplets can take part in Bergeron process, the proportion of those droplets (not in balanced diameter state) in each droplet bin, donated as  $f_i$  (i means the i-th droplet bin), should be added as a correction:

$$\begin{cases} \Delta N_i = -\frac{\Delta R_I}{\Delta L_i} \cdot \frac{R_I}{R_i} \cdot N_i \cdot f_i + \frac{\Delta R_I}{\Delta L_{i+1}} \cdot \frac{R_I}{R_{i+1}} \cdot N_{i+1} \cdot f_{i+1}, i = 1, 2, 3, ..., I - 1; \\ \text{decrease due to evaporation} & \text{increase due to evaporation in next larger bin} \\ \Delta N_I = -\frac{\Delta R_I}{\Delta L_I} \cdot \frac{R_I}{R_I} \cdot N_I \cdot f_I. \end{cases}$$

$$(11)$$

where the definition of  $f_i$  is as follows:

$$f_i = \frac{\text{Number of supercooled droplets containing small CCNs}}{\text{Total number of supercooled droplets in the i-th droplet bin}}$$
(12)

Figure 3 presents a schematic of the size bin concentration for supercooled droplets, taking into account both the WBF and GCCN-ISP. In the third bin of the spectrum, there are 10 droplets—nine with small CCNs and one with a large CCN. Since these supercooled droplets are within the same bin, their curvature effects are nearly identical. However, differences in solute composition result in variations in the solute effect. Consequently,  $f_3$  is 90%, indicating that 90% of the droplets in this bin will participate in the WBF process, while the remaining 10% will coexist with the ice (GCCN-ISP).

#### 175 2.3 Numerical Simulation



Utilizing the theoretical framework outlined above, a forward model is developed in this study. The GCCN-ISP is now illustrated through an example involving a virtual mixed-phase cloud. Suppose we have a mixed-phase cloud characterized by a size bin concentration of supercooled liquid droplets, as depicted in Figure 3. We assume that this cloud is undergoing the WBF process, where liquid water continuously converts to solid ice.

Without delving into the detailed temperature and water vapor conditions of the cloud, we assume for simplicity that the largest supercooled water droplets, initially 49 µm in diameter, are reduced in size by 0.01 µm with each calculation step. Coagulation will be an additional term in Eq. 10 and 11 but we ignore it in this paper because we do not know the "speed" of coagulation versus the "speed" of the WBF process. This process is repeated for a total of 1000 iterations. It's anticipated that as the number of calculations increases towards 1000, the proportion of ice water content within the cloud will gradually approach 100%.

To highlight the significance of the balance diameter in each size bin, we consider three scenarios:

- 1. All supercooled liquid water droplets can participate in the WBF process: In this scenario, the values of f in each size bin is set at 100% for every bin.
- 2. Except for the supercooled water droplets in one arbitrary (e.g. the third) bin, the supercooled water droplets inside the rest of the bins can participate in the WBF process: Here, we focus on the droplets in the third size bin, initially 5 μm in diameter. Over 1000 iterations, these droplets gradually reach a balanced diameter. We set *f* at 70%, 40%, and 10% for successive calculations.
  - 3. Except for the supercooled water droplets in the last bin (the largest size), the supercooled water droplets inside the rest of the bins can participate in the WBF process: Similarly, we examine the droplets in the last size bin, initially 49  $\mu$ m in diameter, which also gradually reach a equilibrium state over 1000 iterations. The f values are set at 70%, 40%, and 10% for successive calculations in this scenario.

Since the effective particle radius/diameter is often used in remote sensing methods to represent the size bin concentration of droplets, the changes in size bin concentration of this mixed phase cloud is reflected by effective particle diameter in the

following study, as given by follows:

$$r_e = \frac{\int\limits_0^\infty \pi \cdot r^3 \cdot n(r) dr}{\int\limits_0^\infty \pi \cdot r^2 \cdot n(r) dr}; D_e = 2 \cdot r_e.$$
 (13)

Where  $r_e$  is the effective radius;  $D_e$  is the effective diameter; n(r) is the size bin concentration of cloud droplet. The results are given in Fig.4, the relationship between the effective diameter and the ice phase fraction coefficient ( $\mu_{ice}$ =IWC/(IWC+LWC)). According to AFLUX observations, particles larger than 50  $\mu$ m are classified as ice (Brown and Francis, 1995). Therefore, the effective particle radius calculated in this study refers specifically to the supercooled liquid droplets (<50  $\mu$ m).

# **3** Verification with Observations




#### 3.1 AFLUX Campaign Observation Data

During the AFLUX field campaign (Mech et al., 2021a) conducted in spring 2019 as part of the Transregional Collaborative Research Centre TR 172 (AC)<sup>3</sup> (Wendisch et al., 2023), the Basler BT-67 research aircraft Polar 5, stationed in Spitzbergen, Norway (78.24 N, 15.49 E), was equipped with advanced in-situ cloud instrumentation by DLR. This instrumentation included the Cloud Aerosol Spectrometer (CAS), Cloud Imaging Probe (CIP), and Precipitation Imaging Probe (PIP), providing comprehensive data on microphysical cloud properties such as particle size bin concentration, total particle number concentration, effective diameter, median volume diameter, and estimated cloud/liquid/ice water content (Moser et al., 2023). The aircraft observations considered here were conducted on March 23, 2019, covering the time period from 11:30 to 17:00 (UTC). The airplane flight path is given in Fig.A1 and the selected mixed phase cloud is indicated by the red markers on the flight path in Fig.A1. During this time, around 13:00, the in-cloud temperature was approximately -14.5  $\pm$ 1.5 °C (Moser and Voigt, 2022). The ice phase fraction coefficient,  $\mu_{ice}$ , is then calculated from the measurement of liquid water content (LWC) and ice water content (IWC),  $\mu_{ice}$ =IWC/(IWC+LWC). The dataset covers particle sizes ranging from 2.8 to 6400.0  $\mu$ m in diameter. The liquid cloud droplets size bin concentration is used to get the effective diameter following Eq.(13). The combined dataset can be downloaded from the pangea repository (Moser and Voigt, 2022).

#### 220 3.2 Backward Trajectories

HYsplit stands for Hybrid Single-Particle Lagrangian Integrated Trajectory Model (Draxler and Hess, 1998; Stein et al., 2015), a widely used atmospheric dispersion model for simulating the transport, dispersion, and trajectory of air pollutants, particles, aerosols, or any atmospheric material released into the atmosphere. Developed by the National Oceanic and Atmospheric Administration (NOAA), HYsplit employs a Lagrangian methodology for tracking the movement of individual air masses or particle trajectories in the atmosphere. Backward trajectory analysis is applied in this study. Specifically, The HYSPLIT model (Version 5.0.0) is employed to perform a 120-hour backward trajectory analysis, initialized at AFLUX campaign flight

locations (81°N, 9°E), using Global Data Assimilation System (GDAS) meteorological data at a resolution of  $1^{\circ} \times 1^{\circ}$  and started with the height on 100 m and 500 m.

#### 4 Results







#### 4.1 Numerical Simulation

Figure 4a illustrates the variation of effective radius with the ice phase fraction coefficient,  $\mu_{ice}$ =IWC/(IWC+LWC). Initially, the black dashed line represents a scenario where the balanced diameter is disregarded, implying that the solute content in each supercooled droplet is low enough to use the formula for the saturated saturation water vapor pressure of pure water droplets. The results indicate a gradual increase followed by a decrease in effective diameter as  $\mu_{ice}$  approaches 1. Ice particle growth is treated through vapor deposition under a temperature-dependent saturation vapor pressure over ice. Detailed growth processes, such as the effect of the ventilation coefficient and mass transfer, are not included, as the focus here is on the phase transition rather than the rate of ice growth.

The black dashed lines in Fig.4b and c represent the variation in the number concentration of small (5  $\mu$ m) and large (49  $\mu$ m) droplets, respectively, as  $\mu_{ice}$  increases without accounting for the GCCN-ISP. Initially, The saturation vapor pressure increases for smaller droplets due to the curvature effect (also known as the Kelvin effect), which enhances evaporation from highly curved droplet surfaces. Therefore, initially, the concentration of small droplets decreases more rapidly than that of large particles due to the higher saturation water vapor pressure of small droplets compared to large ones during the WBF process. Consequently, the loss of liquid water from small droplets occurs at a faster rate, leading to an increase in the percentage of number concentration of large droplets and hence a larger effective diameter. However, as  $\mu_{ice}$  continues to rise, large supercooled droplets are almost depleted, and small droplets dominate again, causing the effective diameter to decrease with increasing  $\mu_{ice}$ .

If we consider a scenario where a portion of small droplets with a diameter of 5  $\mu$ m reach the balanced diameter state, with proportions of those droplets not in balanced diameter state, f set at 70%, 40%, and 10%, the resulting variation in effective diameter is depicted by the blue solid line in Fig.4a for f setting 10% (others are presented in Fig.5). When the proportion of unbalanced particles is 10%, the effective diameter decreases more rapidly and remains lower at the same  $\mu_{ice}$  compared to scenarios with 100% unbalanced particles (compare with the black dashed line in Fig.4a). For higher values of f, such as 40% and 70% (see Fig.5), the effective diameter initially increases and then decreases with increasing  $\mu_{ice}$ , and the decline is less pronounced compared to the f = 10% case. The peak effective diameter achievable decreases with decreasing f, and the rate of decrease accelerates accordingly.

This trend arises because during the WBF process, some small droplets (in this case, 5  $\mu$ m) reach the balanced diameter state, coexisting with ice crystals as long as temperature and ambient water vapor conditions remain unchanged. Consequently, their percentage of number concentration gradually increases with increasing  $\mu_{ice}$ , significantly contributing to the effective diameter, as depicted in Fig.4b. Theoretically, all droplets except those reaching the balanced diameter state will eventually be consumed, resulting in a final effective particle diameter of 5  $\mu$ m.

In contrast, assuming a fraction of large droplets with a diameter of 49  $\mu$ m reach the balanced diameter state, following the same proportions as in the previous experiments with 5  $\mu$ m droplets, yields a different trend. The change in effective particle diameter in this scenario is represented by the solid orange line in Fig.4a. When the proportion of unbalanced particles is 10%, the effective particle diameter increases instead of decreasing compared to scenarios with 100% unbalanced particles. This is attributed to the dominance of the concentration proportion of large droplets reaching the balanced diameter state. The size bin concentration of droplets in this case is shown in Fig.4c, illustrating the increasing proportion with the rise of  $\mu_{ice}$ . Theoretically, the final effective diameter is 49  $\mu$ m when all droplets except those reaching the balanced diameter state are consumed.

Hence, if the effective diameter increases with the ice phase fraction coefficient ( $\mu_{ice}$ ), it indicates that some large droplets have reached the balanced diameter state. Conversely, if the effective diameter initially increases with  $\mu_{ice}$  and then decreases as  $\mu_{ice}$  approaches 1, it suggests that some small droplets are in the balanced diameter state. It's also plausible that each bin of the cloud droplet size bin concentration contains particles in the balanced diameter state, depending on the distribution of f across the cloud droplet size bin concentration. In this case, f is a quantity that follows the distribution of the cloud droplet size bin concentration.

## 4.2 AFLUX Campaign Cloud Observations



290

Figure 6a depicts the effective diameter in measurement, which initially increases and then decreases with the ice phase fraction coefficient ( $\mu_{ice}$ ). This suggests that during the WBF process, either all droplets in this mixed-phase cloud have nonbalanced diameters, or some small droplets are in a balanced diameter state and not undergoing the WBF process. In Figure 6b, we present the decrease in number concentration in each bin as  $\mu_{ice}$  increases from 0.83 to 0.85. The reduction in number concentration calculated theoretically (the black line in Fig.6b) and observed in reality (the blue line in Fig.6b) is generally consistent for all bins except the smallest one. For this bin, the observed reduction is approximately  $-2.0 \times 10^5$  m<sup>-3</sup>, which 280 is less than the theoretically calculated value of approximately  $-2.0 \times 10^6 \text{ m}^{-3}$ . This discrepancy suggests that about 13.3% of the droplets are not in equilibrium state (f), leaving around 86.7% of droplets with a diameter of 5.09  $\mu$ m in a balanced diameter state. After consideration of those droplets not in balanced diameter, f = 13.3%, there is a good agreement between the corrected reduction in number concentration calculated theoretically (the red dashed line in Fig.6b) and that observed in reality. Compared with the data presented in Figure 2, considering a temperature of  $-14.5^{\circ}$ C and a droplet diameter of 5.09 μm, the dry cloud droplet condensation nucleus diameter is estimated to be around 2 μm. As the supercooled droplets are nearly depleted (i.e., as  $\mu_{ice}$  approaches 1.0), the observed particle size bin concentration shows that the bin with the highest number of remaining particles falls between 2.5 and 5 µm in diameter, as shown in Fig.7. This result aligns with our expectations.

During the AFLUX campaign, a 5-day backward trajectory analysis reveals that the air mass at 100 m altitude originated from the central Arctic region, while the air mass at 500 m altitude was transported from the North Atlantic, passing over the southwestern part of Greenland, as shown in Fig. 8. The trajectory starting at 500 m altitude indicates that the air mass initially traveled along the southeastern coast of Greenland before reaching Ny-Ålesund, predominantly remaining below 100 m altitude and over ice-covered regions. This pathway suggests that the cloud condensation nuclei in the mixed-phase cloud

observed during the AFLUX campaign likely originated from this region and could have been predominantly composed of sea salt aerosols (probably generated by blowing snow/frost flowers).

## 4.3 Summary of Results





The existence of the GCCN-ISP has been theoretically derived and compared to in-situ observations. This process appears to be less influenced by the initial state of liquid droplet size bin concentration and more by the temperature of the cloud and the solute concentration within individual droplets. Moreover, according to the GCCN-ISP, as a mixed-phase cloud evolves and the supercooled liquid water content gradually diminishes, the droplets that remain coexist with ice at this stage could be predominantly those nucleated on giant CCNs. Therefore, sampling droplets for direct observation of their solute mass during the final stage of the mixing cloud would be a more intuitive validation of the process. Additionally, laboratory experiments using cloud chambers or particle traps could provide controlled conditions to better understand microphysical processes like GCCN-ISP and its interaction with the WBF effect.

#### 5 Discussion

## 5.1 Can the WBF Process Coexist with GCCN-ISP?

As shown in Fig.1b, for all supercooled water droplets with a diameter of 20  $\mu$ m, the behavior of droplets depends on the size bin concentration of the CCN they contain (ranging from 1 to 10  $\mu$ m) under an ambient temperature of -10°C. Supercooled droplets containing smaller CCN, such as those with a size of 5  $\mu$ m, exhibit a saturation vapor pressure higher than that of the ice surface, causing them to lose water, which is transferred to ice crystals. In contrast, droplets with larger CCN, such as 10  $\mu$ m, have a saturation vapor pressure lower than that of the ice surface (0 > Temperature > -17°C) and compete with ice crystals for water vapor. Therefore, the WBF process and the GCCN-ISP process could probably coexist.

The coexistence of these two mechanisms makes it possible for some mixed-phase clouds to have longer lifetimes. In WBF condition, relative humidity with respect to water typically remains below saturation, whereas it is above saturation relative to ice. This difference leads to rapid evaporation of supercooled droplets, typically on timescales of minutes to tens of minutes, rapidly transitioning mixed-phase clouds toward an ice cloud (Morrison et al., 2012; Korolev et al., 2020). However, when GCCN-ISP conditions coexist, supercooled droplets containing giant CCN can maintain equilibrium at humidity levels close to ice saturation, thus limiting their participation in the WBF process. Consequently, a stable population of supercooled droplets persists, prolonging the lifetime of mixed-phase cloud.

To understand the WBF process and GCCN-ISP, Fig.9 shows a schematic of the variation of saturated saturation water vapor pressure with temperature and solution solute. There are four shaded area in this figure, and area B is the region with WBF process typically. In area C, both ice particles and liquid droplets grow; In area A, both ice particles and droplets evaporate. On the other hand, if the solute of the droplet is gradually increasing, that is, the aerosol is getting larger in size, the saturated saturation water vapor pressure over this droplet will gradually decrease, as shown by the yellow arrow in this figure. Hence,

the area B or area A will be divided into two parts then area D appears. In this figure, the aforementioned area A becomes the area where ice particles decreases while droplets increase, and in area D, both ice particles and droplets evaporate. Similarly, the B area will be divided into two parts in the same way: the upper region where the GCCN-ISP occurs, and the lower region where the WBF process occurs. The blue curve in this figure is drawn arbitrarily, to make it easier to distinguish different regions. In reality, the blue curve should be calculated based on actual the droplet and CCN sizes, as done in Fig.1a, to define the regions more accurately. The key parameter determines in which region is how much aerosol is actually inside the droplet. Thus, the two processes could eventually co-exist in the same cloud of mixed states as long as the mass of solute inside the droplets is not the same.

#### **5.2** Where do These Coarse Aerosols Come from?





In the Arctic area, the size of aerosols is often very tiny, smaller than 1 µm (Boyer et al., 2023). Generally on the Earth, small aerosols, such as biological aerosol, are important for both INP and CCN (Mikhailov et al., 2021). The key point in the GCCN-ISP is where those coarse aerosols, e.g. larger than 10 µm, come from. Recently, some observations show that the coarse sea salt aerosols can be emitted from the blowing snow during winter time. The size of sea salt aerosol can reach to 10 µm in diameter (Huang and Jaeglé, 2016; Yang et al., 2018; Frey et al., 2020; Gonzalez et al., 2023). Thus, primary sea salt aerosols emitted directly from blowing snow or surrounding open waters may be a potential natural source of coarse aerosols to the GCCN-ISP. If so, there will be more mixed phase clouds in the oceans than on land.

A second possible source is the existence of a mechanism by which aerosols are continuously enriched within the droplet, growing from small to large. Generally, in the clear Arctic, even very small aerosol acts as CCN. Jung et al. (2018) show that at 1% supersaturation (RH = 101%) in winter almost all and in summer at least 40% of all aerosol >10 nm grow into CCN in the Arctic. In addition, Hoffmann and Feingold (2023) demonstrate that large-scale dynamics enable aerosol particles to undergo a cycle of droplet condensation, collision-coalescence, and evaporation, resulting in a gradual increase in aerosol size in warm clouds. In mixed-phase clouds where WBF process dominates, collisions and condensation of supercooled liquid water droplets are not important. The main process that forms cold clouds is the freezing process (Shaw and Lamb, 1999; Drdla et al., 2002; Bacer et al., 2021). When a supercooled liquid droplet collides with another supercooled liquid droplet (Alkezweeny, 1969) or comes in contact with a solid surface, the supersaturated droplet gradually freezes and forms ice crystal particles (Jung et al., 2012; Shayunusov et al., 2021). Based on that, if the tiny aerosol is taken for granted in mixed phase cloud, it means the supercooled liquid water droplets will be in a very narrow distribution of droplet size, which doesn't fit the observations. Many observations have shown that the size of supercooled water droplets can vary greatly (Zhao and Lei, 2014; Mioche, 2017). Therefore, the size of a supercooled water droplet in mixed phase cloud should not only grow by condensation of water molecules, but have a similar collision-coalescence process (Young et al., 2016; Braga et al., 2025) in warm cloud. When salt water freezes, the concentration of solutes in the solution becomes higher (Ginot et al., 2020), as ice crystals cannot incorporate most alien molecules (Panday and Corapcioglu, 1991). James et al. (2021) show that secondary drops were generated during both the spreading and retraction phase of the supercooled water drop impact. Therefore, we propose that the second possible cause of coarse aerosol particles is the collision between supercooled water droplets. During these collisions,

some of the supercooled water droplets freeze into ice crystals, while others continue to exist as secondary water droplets. As these secondary water droplets collide, their solute concentration increases, leading to the formation of coarse mode aerosol particles.

Therefore, in future work, we aim to deepen our understanding of cloud dynamics by focusing on clouds initially composed of small dry radius particles, typically around 0.1 µm. Our approach will involve a detailed analysis of particle size changes resulting from coagulation and the turbulent influx of water vapor from above and below the cloud layer.

# 6 Conclusions






In conclusion, the Wegener-Bergeron-Findeisen (WBF) process, a cornerstone of cloud physics, describes the transition between supercooled liquid water and ice crystals within a specific temperature range. In this study, the GCCN-ISP is found, where the presence of coarse-core cloud condensation nuclei can compete with ice crystal particles for moisture in the air or even liquid water in supercooled water droplets. Those supercooled water droplets with coarse aerosol can coexist with ice crystals in mixed-phase clouds when they reached the balanced diameters. The theoretical framework developed here elucidates this process, which has been validated through in-situ observations from aircraft. Finally, the coarse aerosol need to be taken into account carefully, and the GCCN-ISP might be helpful in the formation of mixed phase clouds, potentially extending the lifetime of mixed-phase clouds and thereby impacting the energy balance of the climate system.

To validate and expand our understanding of the GCCN-ISP, further observations and experimental efforts are essential. Field campaigns employing research aircraft equipped with cloud probes, will provide high-resolution data on cloud properties. Beyond the AFLUX campaign, several other aircraft-based field studies have investigated cloud—aerosol interactions and the microphysical properties of mixed-phase clouds across a range of geographic regions. For example, the ACTIVATE (Aerosol Cloud McTeorology Interactions over the Western Atlantic Experiment) campaign, conducted by NASA over the U.S. East Coast and western North Atlantic, has provided high-resolution measurements of CCN size distribution (0.1 - 2.6 µm), aerosol composition, and cloud microphysics using dual-aircraft observations (Sorooshian et al., 2025). Similarly, the NAAMES (North Atlantic Aerosols and Marine Ecosystems Study) project included multiple aircraft missions over the North Atlantic to characterize CCN variability (0.02 - 0.5 µm) and its link to seasonal biogenic emissions (Behrenfeld et al., 2019). Additionally, direct measurements of CCN size and composition (e.g., using aerosol mass spectrometers), when available, offer important means to corroborate inferred CCN properties. Together, these observational efforts highlight the broader relevance of large-particle activation mechanisms and offer valuable datasets for testing the applicability of the GCCN-ISP process under diverse regional and seasonal conditions. Satellite missions with capabilities to detect mixed-phase clouds, such as the EarthCARE satellite, can offer comprehensive global coverage. High-resolution climate

While our study focuses on the saturation vapor over supercooled droplets with large CCNs, the ice-phase processes in mixed-phase clouds are inherently complex. Secondary ice production (SIP) mechanisms, such as the Hallett–Mossop process, are generally restricted to a narrow temperature range of -3°C to -8°C (Korolev and Leisner, 2020). The cloud temperature analyzed in this study is around -14.5°C, making SIP unlikely. With this consideration, in the current modeling framework, the

SIP process is excluded. Extending this work with more comprehensive models that incorporate detailed cloud microphysics, combined with long-term monitoring programs and extensive data analysis, will help identify the ice particle shape, ventilation effects, aggregation, and turbulence will be important for assessing the robustness of this mechanism under realistic conditions.

Beyond numerical modeling, laboratory validation is critical. Cloud chamber facilities such as AIDA (Aerosol Interaction and Dynamics in the Atmosphere (Lamb et al., 2023)) and the Pi Chamber (Wang et al., 2024) offer controlled environments to systematically vary CCN size, composition, and ambient conditions, allowing detailed observation of droplet—ice competition. While such platforms have been extensively used for studying ice nucleation and SIP, targeted experiments focusing on solute-induced suppression of ice growth remain limited. We encourage future laboratory activities that replicate conditions relevant to the Arctic to explore solute-driven persistence in supercooled droplets.

Finally, we emphasize the broader climatic relevance of the GCCN-ISP mechanism. The persistence of supercooled liquid water affects the radiative properties and lifetime of mixed-phase clouds. Suppressing the WBF process via solute effects could extend cloud persistence, potentially altering cloud radiative forcing and Arctic amplification. This mechanism also highlights the need to better represent giant CCN and their solute effects in models, as current schemes tend to underestimate their influence on phase partitioning. Although the present study does not explicitly quantify such impacts, we hope to explore these implications in future work by coupling the GCCN-ISP framework with climate models.

Data availability. The can AFLUX campaign be downloaded from: https://doi.org/10.1594/PANGAEA.940564 (Moser and Voigt, 2022). The airborne observational datasets are provided by Mech et al. (2022). The datasets from the AFLUX (Mech et al., 2021a) and MOSAiC-ACA (Mech et al., 2022) campaigns are publicly available through the PANGAEA database.

Author contributions. D.J. developed the forward model, conducted data processing, and wrote the article. C.R. and X.S. provided valuable assistance in refining the forward model, contributing to the theoretical aspects, and supplying observational data. M.M and C.V. provided data support for this paper from aircraft observations. M.P. and J.N. offered important feedback on the article's review and helped improve its structure. All authors commented on this manuscript and contributed to the article's review.

Competing interests. The authors declare no competing interests.


Acknowledgements. We gratefully acknowledge the funding by the Deutsche Forschungsgemeinschaft (DFG, German Research Foundation)

– project number 268020496 – TRR 172, within the Transregional Collaborative Research Center "ArctiC Amplification: Climate Relevant Atmospheric and SurfaCe Processes, and Feedback Mechanisms (AC)3". For airborne campaigns, we thank the work from Mech et al. (2022). Collections of the datasets have been compiled and can be found on the public database PANGAEA for both campaigns, AFLUX (Mech et al., 2021a) and MOSAiC-ACA (Mech et al., 2021b). We thank the senate of Bremen for partial funding of this work. In addition,

| 120 | we would like to give special thanks to Prof. Dr. Manfred Wendisch, Dr. Marcus Klingebiel, Dr. Christof Lüpkes, and Dr. Ulrike Wacker, who provided important feedback and suggestions to improve our study. |  |
|-----|--------------------------------------------------------------------------------------------------------------------------------------------------------------------------------------------------------------|--|
|     |                                                                                                                                                                                                              |  |
|     |                                                                                                                                                                                                              |  |
|     |                                                                                                                                                                                                              |  |
|     |                                                                                                                                                                                                              |  |
|     |                                                                                                                                                                                                              |  |
|     |                                                                                                                                                                                                              |  |
|     |                                                                                                                                                                                                              |  |
|     |                                                                                                                                                                                                              |  |
|     |                                                                                                                                                                                                              |  |
|     |                                                                                                                                                                                                              |  |

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

**Figure 1.** (a) Water vapor pressure over liquid water surface (red line), ice surface (black line), and various sizes of CCN particles in droplets, plotted as functions of temperature. CCN particles radius are 0.1 (blue dashed lines with circles), 1 (pink line), 5 (pink dashed line), and 10 μm (blue line), respectively, while the cloud droplet radius is fixed at 20 μm for all cases. (b) The difference in water vapor pressure between liquid water/supercooled liquid droplets and ice surfaces, plotted as a function of temperature. The Legend for (b) is the same as in (a). The yellow cross mark is the temperature required to equalize the saturated saturation water vapor pressure of the supercooled water droplet with CCN size of 10 μm and the ice surface.

**Figure 2.** (a)The relationship between the diameter of dry salt particles (NaCl) and the corresponding balanced diameter of droplets. Each black line represents the diameter of the dry salt particles; (b) Ratio of droplet diameter in equilibrium state to dry aerosol diameter for different aerosol compositions.

**Figure 3.** A schematic of the size bin concentration for supercooled droplets, taking into account both the WBF and CCN-ISP. The arrows indicate the direction of water transfer in a mixed phase cloud.

Figure 4. (a) The relationship between the effective diameter and the ice phase fraction coefficient,  $\mu_{ice}$ . The proportion of those droplets (not in balanced diameter state) is given as f. The black line represents the scenario where all droplets are in an unbalanced diameter state, which means all supercooled liquid water droplets can participate in the WBF process, f is 100% (compare sec.2.3, first scenario). The blue lines represent scenarios where a portion of small droplets (5  $\mu$ m) are in a balanced diameter state, with proportions of 10% (cases of 40% and 70% are given in Fig.5). The orange lines represent scenarios where a portion of large droplets (49  $\mu$ m) are in a balanced diameter state, with proportions of 10% (cases of 40% and 70% are given in Fig.5). (b) Variation in the number of droplets in the third bin of the droplet size bin concentration as a function of ice phase fraction coefficients. (c) depicts similar information as (b), but for largest bin of the cloud size bin concentration . All figures share the line labeling as in Fig.4 (a).

Figure 5. The relationship between the effective diameter and the ice phase fraction coefficient,  $\mu_{ice}$ . The proportion of those droplets (not in balanced diameter state) is given as f. The black line represents the scenario where all droplets are in an unbalanced diameter state, which means all supercooled liquid water droplets can participate in the WBF process, f is 100% (compare sec.2.3, first scenario). The blue lines represent scenarios where a portion of small droplets (5  $\mu$ m) are in a balanced diameter state, with proportions of 10%, 40% and 70%, respectively. The orange lines represent scenarios where a portion of large droplets (49  $\mu$ m) are in a balanced diameter state, with proportions of 10%, 40% and 70%, respectively.

Figure 6. (a) The relationship between the effective diameter measured from AFLUX campaign and the ice phase fraction coefficient,  $\mu_{ice}$ . (b) The decreased number concentration of each bin as  $\mu_{ice}$  increases from 0.83 to 0.85 in the calculation without f correction (black line), the observation (the blue line) and the calculation with f=13.3% correction (red dashed line). The negative values on the Y-axis indicate the reduction in the number of supercooled droplets at each bin as the  $\mu_{ice}$  increases (from 0.83 to 0.85). Note that we filtered the data using a threshold of cloud water content (CWC)> 0.01 g  $m^{-3}$  to ensure that only cloud-containing measurements were included.

Figure 7. The particle number concentration of each bin from AFLUX campaign in-situ measurement when the ice phase fraction coefficient,  $\mu_{ice}$ , approaches 1.0.

**Figure 8.** The 5-day backward trajectory of the mixed-phase cloud in the red-marked region in Fig.A1. Dots represent starting points at 100 m altitude, while triangles represent starting points at 500 m altitude. Different colors indicate changing altitudes along the backward trajectory.

**Figure 9.** The Water vapor pressure over liquid water surface (red line), ice surface (black line), and water vapor pressure with coarse aerosol (blue line). The blue curve in the figure is drawn arbitrarily, to make it easier to distinguish different regions. In reality, the blue curve should be calculated based on actual the droplet and CCN sizes, as done in Fig.1a, to define the regions more accurately.

Figure A1. Plot of an airplane flight path, highlighting the selected mixed-phase cloud region in red, with temperatures ranging -14.5  $\pm 1.5$  °C.