# Peer review of "Giant Cloud Condensation Nuclei enhanced Ice Sublimation Process: A potential mechanism in mixed phase clouds"

_EGUsphere, 2025_

## Referee Comment (RC1)

Review of "Giant Cloud Condensation Nuclei enhanced Ice Sublimation Process: A potential mechanism in mixed phase clouds", by Ji et al., egusphere-2025-1932.

Many Cloud Physicists and of course Chemists know that the relative humidity with respect to water (RH) over a pure sodium chloride flat surface is about 76% and ammonium sulfate is about 80% (see table below). Due to the Kelvin effect, the RH over a spherical pure sodium chloride solution is higher. The Kelvin effect becomes more important as aerosol particle diameter decreases due to the increasing curvature of the aerosol surface which impacts the saturation vapor pressure of aerosol constituents at the surface. Therefore, the larger the pure salt sodium chloride cloud condensation nuclei, the lower the saturation vapor pressure over the resulting particle surface; the equilibrium RH is calculable. The effect is represented by Eq. (3) in the article.

Figure 1 shows pictorially how the vapor pressure over a salt solution droplet of different sizes compares to the saturation vapor pressure over ice, and where the crossover between the vapor pressure over the solution droplets becomes less than the vapor pressure over an ice surface.

In this study, the authors have clearly demonstrated the potential effect whereby droplets containing giant salt CCN can sublimate ice crystals.

| Saturated Salt Solution | Temperature (°C) | | | | | | | | | | |
|---|---|---|---|---|---|---|---|---|---|---|---|
| | 0 | 5 | 10 | 15 | 20 | 25 | 30 | 35 | 40 | 50 | 60 |
| | Relative Humidity over the Salt Solution (%) | | | | | | | | | | |
| Ammonium sulphate - $(NH_4)_2SO_4$ | 82 | 82 | 82 | 82 | 81 | 81 | 81 | 80 | 80 | 79 | |
| Sodium chloride - NaCl | 76 | 76 | 76 | 76 | 75 | 75 | 75 | 75 | 75 | 74 | 75 |

The authors have used a model and data from the AFLUX campaign to determine the CCN size/droplet size under which the Giant Cloud Condensation Nuclei-Enhanced Ice Sublimation Process (GCCN-ISP) can be operative.

How are the giant CCN produced in high latitude regions such as Svalbard where the AFLUX measurements were made? The authors propose two processes. Primary sea salt aerosols emitted directly from blowing snow or surrounding open waters. A second possible process to cause these giant CCN particles is the collision between supercooled water droplets.

Primary Comments

2.1 and elsewhere saturated to saturation

Lines 124-125. You mention no new water supplied by turbulent mixing. First, you mean by updrafts. Second, you should discuss the implications of having updrafts on your model calculations.

125. decreases at a rate
Should be changes at a rate as follows.

125. Mention that S is dependent upon the droplet/ice particle size distributions and vertical velocity.

127-128. It also changes due to S. You assume that S decreases.

177-186 What if vertical velocities of let's say 10, 50 and 100 cm/s are assumed

225. What you mean here is the curvature effect

4.1 Numerical Simulation How is the growth of the ice particles considered. That is, things like ventilation coefficient, particle mass, etc

271 particle size distribution. How about the ice particle size distribution, which is interesting and relevant. Also, effective diameter usually considers both the liquid and ice phase together

What are the implications of the proposed process?

Minor Comments

100 Combining

196 Spitzbergen, Norway

---

## Author Comment (AC1)

Response to Reviewer #1

We sincerely thank Reviewer #1 for the constructive and insightful feedback. We appreciate the recognition of the conceptual advance provided by the GCCN-ISP mechanism, and we address each suggestion below (black text: reviewer's comments, blue text: our response, orange text: manuscript revisions):
* * *
Comment 1 (Line 102 and elsewhere): "'saturated' should be changed to 'saturation'

Response: Corrected. And we have revised the manuscript to replace all incorrect uses of "saturated vapor pressure" with "saturation vapor pressure" to ensure consistency and accuracy.
* * *
Comment 2 (Lines 124–125): "You mention no new water supplied by turbulent mixing. First, you mean by updrafts. Second, you should discuss the implications of having updrafts on your model calculations."

Response: We thank the reviewer for this helpful comment and have revised the manuscript accordingly.

In atmospheric clouds, the evolution of cloud droplet size is influenced by two key factors: (1) the ambient water vapor pressure, which controls the vapor availability for condensation or deposition, and (2) the equilibrium saturation vapor pressure over the droplet surface, which depends on droplet size, curvature, and composition (via the solute effect).

Vertical motion (updrafts) affects the ambient vapor pressure by inducing adiabatic cooling and enhancing supersaturation (G.Abade et al., 2024; M.Guti'errez et al., 2023). Such updrafts can slow the Wegener–Bergeron–Findeisen (WBF) process by supplying moisture and sustaining supersaturation, thereby delaying the depletion of supercooled droplets (G.Abade et al., 2024; A.Khain, M. Pinsky and A. Korolev., 2021; A. Korolev and I. Mazin., 2003). Vertical motion determines the ambient water vapor pressure, thereby affecting the activation of cloud droplets and the rate of primary ice formation. This dynamical factor operates independently from the solute effect of cloud condensation nuclei (CCN), which modifies the equilibrium properties of individual droplets. In contrast, the solute effect in our study, acts by lowering the equilibrium saturation vapor pressure over supercooled droplets, thereby changing the thermodynamic balance point. It can allow liquid droplets to persist even when the environment would otherwise favor ice growth. In this way, both updrafts and solute effects can extend the lifetime of mixed-phase clouds—but via distinct physical mechanisms.

This distinction motivates our modeling choice: by neglecting vertical motion, we isolate the solute effect as an internal microphysical process, enabling a clear quantification of its standalone contribution to droplet–ice vapor competition. We have added the following discussion:

Line 129 "As mentioned earlier, in this equation, there is no new water supplied by turbulent mixing, meaning that updrafts are not considered. In atmospheric clouds, the evolution of cloud droplet size is governed by two key factors: (1) the ambient water vapor pressure, which determines the vapor availability for condensation or deposition, and (2) the equilibrium saturation vapor pressure over the droplet surface, which is influenced by droplet size, curvature, and composition (via the solute effect). Updrafts affect cloud microphysics by cooling air parcels and enhancing supersaturation, thereby suppressing the WBF process and delaying ice growth (G.Abade et al., 2024; M.Guti'errez et al., 2023; M. Pinsky and A. Korolev., 2021). In contrast, the solute effect reduces the surface saturation vapor pressure, directly influencing whether supercooled droplets can persist. This study intentionally excludes vertical motion to isolate the microphysical contribution of the solute effect to droplet–ice vapor competition."
* * *
Comment 3 (Line 125): "'decreases at a rate' should be changed to 'changes at a rate as follows'"

Response: We have revised the sentence to "changes at a rate as follows" to allow both increases and decreases of supersaturation, depending on the conditions.
* * *
Comment 4 (Line 125): "Mention that S is dependent upon the droplet/ice particle size distributions and vertical velocity."

Response: We have revised the text to explicitly state that the supersaturation is influenced by both droplet and ice particle size distributions, as well as vertical velocity, which together determine the rates of vapor condensation and deposition. As our response in Comment 2, the discussion of the vertical motion effects has been added to the revised manuscript.
* * *
Comment 5 (Lines 127–128): "It also changes due to S. You assume that S decreases."

Response: We agree and correct it in the text.
* * *
Comment 6 (Lines 177–186): "What if vertical velocities of let's say 10, 50 and 100 cm/s are assumed?"

Response: Thank you for this suggestion. As noted in our response to Comment 2, we intentionally neglect vertical motion in our model to isolate the microphysical role of solute effects on the droplet–ice vapor balance. Including vertical velocity would modify the ambient water vapor pressure and enhance supersaturation, thereby influencing the rate of droplet activation and primary ice formation.

Previous studies have investigated this aspect in detail. For example, Lu et al. (2012) show that with increasing vertical velocity the droplet number concentration increases while the relative dispersion decreases. Specifically, the study by J. Bühl, P. Seifert, R. Engelmann and A. Ansmann. (2019) demonstrates that increasing the standard deviation of vertical velocity from 0.1 to 1.0 m/s leads to a two-fold increase in the mass flux of ice water in clouds with cloud-top temperatures below −12°C.

We acknowledge its importance while emphasizing that the goal of our work is to provide a theoretical baseline under specific conditions. We agree that applying prescribed updrafts (e.g., 10–100 cm/s) in future model extensions will help quantify how solute effects and dynamic forcing interact.
* * *
Comment 7 (Line 225): "What you mean here is the curvature effect."

Response: We have revised the manuscript to explicitly name the curvature effect (Kelvin effect), which influences the equilibrium RH of droplets depending on size.

Line 239: The saturation vapor pressure increases for smaller droplets due to the curvature effect (also known as the Kelvin effect), which enhances evaporation from highly curved droplet surfaces.
* * *
Comment 8 (Section 4.1): "How is the growth of the ice particles considered? That is, things like ventilation coefficient, particle mass, etc."

Response: Thank you for pointing this out. In our model, we do not explicitly calculate the growth rate of ice particles, as the focus is on the phase transition from supercooled liquid to ice rather than on the detailed microphysics of ice growth. We do not include the ventilation coefficient, particle mass, or other factors that influence the precise growth rate of ice particles. These processes, such as riming or aggregation, depend on microphysical details beyond the scope of this study. We intend to capture the onset and general behavior of freezing, rather than to quantitatively model the full evolution of ice particle size.We have clarified this modeling choice in Section 4.1:

Line 234: "Ice particle growth is treated through vapor deposition under a temperature-dependent saturation vapor pressure over ice. Detailed growth processes, such as the effect of the ventilation coefficient and mass transfer, are not included, as the focus here is on the phase transition rather than the rate of ice growth."
* * *
Comment 9 (Line 271): "Particle size distribution. How about the ice particle size distribution, which is interesting and relevant. Also, effective diameter usually considers both the liquid and ice phase together."

Response: In this study, we focus on the size distribution of supercooled liquid droplets, and the effective radius calculated refers exclusively to the liquid phase. Following the classification criteria used in AFLUX measurements and previous studies (e.g., Brown and Francis, 1995), particles smaller than 50 μm are assumed to be droplets, while those larger than 50 μm are classified as ice.

We have now clarified this point in the manuscript by explicitly stating that the calculated effective radius corresponds only to the droplet mode, not the combined liquid–ice population.

Line 203 "According to AFLUX observations, particles >50 μm are classified as ice (Brown and Francis, 1995). Therefore, the effective particle radius calculated in this study refers specifically to the supercooled liquid droplets (<50 μm)."
* * *
Comment 10: "What are the implications of the proposed process?"

Response: We thank the reviewer for this important question. Many current models assume CCNs are primarily small particles (<1 μm), which tends to result in rapid ice growth via the WBF process and an underestimation of cloud liquid water content and lifetime. Our study highlights that the presence of giant CCN can enhance local vapor depletion around large droplets, thus prolonging liquid water persistence. This finding may help explain the observed longevity of mixed-phase clouds in the Arctic and underscores the need to incorporate GCCN-ISP into atmospheric models to improve simulation of cloud phase partitioning and climate feedbacks.

Line 400 "Finally, we emphasize the broader climatic relevance of the GCCN-ISP mechanism. The persistence of supercooled liquid water affects the radiative properties and lifetime of mixed-phase clouds. Suppressing the WBF process via solute effects could extend cloud persistence, potentially altering cloud radiative forcing and Arctic amplification. This mechanism also highlights the need to better represent giant CCN and their solute effects in models, as current schemes tend to underestimate their influence on phase partitioning. Although the present study does not explicitly quantify such impacts, we hope to explore these implications in future work by coupling the GCCN-ISP framework with climate models."
* * *
References:

G. Abade et al. "Persistent mixed-phase states in adiabatic cloud parcels under idealised conditions." Quarterly Journal of the Royal Meteorological Society, 150 (2024): 3450 - 3474. https://doi.org/10.1002/qj.4775.

M. Guti'errez et al. "Steady-state supersaturation distributions for clouds under turbulent forcing." Journal of the Atmospheric Sciences (2023). https://doi.org/10.1175/jas-d-23-0155.1.

A. Khain, M. Pinsky and A. Korolev. "Combined effect of the Weber-Bergeron-Findeisen mechanism and large eddies on microphysics of mixed-phase stratiform clouds." Journal of the Atmospheric Sciences (2021). https://doi.org/10.1175/jas-d-20-0269.1.

A. Korolev and I. Mazin. "Supersaturation of Water Vapor in Clouds." Journal of the Atmospheric Sciences, 60 (2003): 2957-2974. https://doi.org/10.1175/15200469(2003)060<2957:SOWVIC>2.0.CO;2.

J. Bühl, P. Seifert, R. Engelmann and A. Ansmann. "Impact of vertical air motions on ice formation rate in mixed-phase cloud layers." npj Climate and Atmospheric Science, 2 (2019): 1-5. https://doi.org/10.1038/s41612-019-0092-6.

Lu, C., Y. Liu, S. Niu, and A. M. Vogelmann (2012), Observed impacts of vertical velocity on cloud microphysics and implications for aerosol indirect effects, Geophys. Res. Lett., 39, L21808, doi:10.1029/2012GL053599.

E. Ávila, R. Pereyra, N. Castellano and C. Saunders. "Ventilation coefficients for cylindrical collectors growing by riming as a function of the cloud droplet spectra." Atmospheric Research, 57 (2001): 139-150. https://doi.org/10.1016/S0169-8095(01)00067-9.

Wusheng Ji and Pao K. Wang. "Ventilation coefficients for falling ice crystals in the atmosphere at low-intermediate Reynolds numbers." Journal of the Atmospheric Sciences, 56 (1999): 829-836. https://doi.org/10.1175/1520-0469(1999)056<0829:VCFFIC>2.0.CO;2.

Brown, P. R. A., and P. N. Francis, 1995: Improved Measurements of the Ice Water Content in Cirrus Using a Total-Water Probe. J. Atmos. Oceanic Technol., 12, 410–414, https://doi.org/10.1175/1520-0426(1995)012<0410:IMOTIW>2.0.CO;2.

---

## Author Comment (AC2)

Response to Reviewer #2

We sincerely thank Reviewer #2 for the constructive and insightful feedback. We appreciate the recognition of the conceptual advance provided by the GCCN-ISP mechanism, and we address each suggestion below (black text: reviewer's comments, blue text: our response, orange text: manuscript revisions):

1. Generalizability beyond the AFLUX campaign

Comment: The reliance on a single case study (AFLUX campaign) limits the generalizability of GCCN-ISP. Expanding validation to diverse geographic regions (e.g., mid-latitude mixed-phase clouds) and seasons would strengthen the conclusions. Direct measurements of CCN size/composition (e.g., via aerosol mass spectrometers), if available, can be used to corroborate inferred CCN properties.

Response: We agree that broader validation across seasons and regions is essential. The AFLUX campaign was selected for its detailed in situ measurements and well-characterized Arctic mixed-phase clouds, which provide a valuable starting point. We added other campaigns with similar measurements of cloud properties and possible future applications in the revised manuscript:

Manuscript revision (Line 376):
"Beyond the AFLUX campaign, several other aircraft-based field studies have investigated cloud–aerosol interactions and the microphysical properties of mixed-phase clouds across a range of geographic regions. For example, the ACTIVATE (Aerosol Cloud MeTeorology Interactions over the Western Atlantic Experiment) campaign, conducted by NASA over the U.S. East Coast and western North Atlantic, has provided high-resolution measurements of CCN size distribution (0.1 - 2.6 μm), aerosol composition, and cloud microphysics using dual-aircraft observations  (Sorooshian et al., 2025). Similarly, the NAAMES (North Atlantic Aerosols and Marine Ecosystems Study) project included multiple aircraft missions over the North Atlantic to characterize CCN variability (0.02 - 0.5 μm) and its link to seasonal biogenic emissions (Behrenfeld et al., 2019). Additionally, direct measurements of CCN size and composition (e.g., using aerosol mass spectrometers), when available, offer important means to corroborate inferred CCN properties. Together, these observational efforts highlight the broader relevance of large-particle activation mechanisms and offer valuable datasets for testing the applicability of the GCCN-ISP process under diverse regional and seasonal conditions."

2. Simplified aerosol composition ($\kappa$ = 1.4 for NaCl)

Comment: The assumption of pure NaCl CCN ($\kappa$ = 1.4) oversimplifies real-world aerosol diversity. Incorporating mixed CCN types (e.g., organics, sulfates) and non-spherical shapes would enhance realism.

Response: Thank you for raising this important point. Indeed, the hygroscopicity parameter $\kappa$ can vary depending on aerosol composition, and assuming a constant value of $\kappa$ = 1.4 (pure NaCl) does not reflect the full diversity of atmospheric particles. However, this simplification is made for two reasons.

First, as discussed in the manuscript, large CCN particles (with dry diameters >1 μm) in the Arctic are most likely dominated by sea salt, especially during winter and early spring. Therefore, we used NaCl as a physically reasonable assumption to construct the look-up table for saturation vapor pressure reduction.

Second, to examine the sensitivity of the process to composition, we also present a comparison using sulfate ($\kappa$ ~ 0.8) in Figure 2b. This allows us to show that while the magnitude of the vapor pressure reduction differs slightly, the overall structure and trends in the equilibrium behavior remain consistent between NaCl and sulfate.

We agree that if the internal composition of CCN within individual droplets could be measured, then a more accurate κ-based correction could be applied. This would improve the quantitative accuracy of the vapor equilibrium calculations. We recommend future work to incorporate a realistic distribution of CCN hygroscopicities and shapes to better represent ambient conditions.

3. Dynamic processes not considered (e.g., turbulence, entrainment)

Comment: The model neglects dynamic processes like turbulence and entrainment, which could modulate GCCN-ISP efficiency.

Response: We acknowledge that our current work does not explicitly include dynamic processes such as turbulence, entrainment, or localized updrafts, all of which are known to affect cloud microphysical evolution.

In particular, dynamic processes (e.g. vertical motions) influence the ambient water vapor pressure through adiabatic cooling, thereby increasing supersaturation and delaying the depletion of supercooled droplets. This effect, which modulates the efficiency of the Wegener–Bergeron–Findeisen (WBF) process, has been extensively documented in previous studies (e.g., Abade et al., 2024; Gutierrez et al., 2023; Pinsky and Korolev, 2021). Turbulent mixing and entrainment further shape the spatial heterogeneity of supersaturation fields, impacting both the activation and the competition between cloud droplets and ice particles.

In contrast, the focus of our study is to isolate the solute effect—an internal microphysical mechanism that reduces the equilibrium saturation vapor pressure over supercooled droplets. This mechanism enables liquid droplets to persist even in conditions where the environment would otherwise favor ice growth.

To clearly quantify the standalone impact of the solute effect, we intentionally omit dynamic forcing such as vertical motion or turbulence. Nevertheless, we agree that future modeling work, particularly with cloud parcel models or large eddy simulations (LES), should incorporate dynamic processes to evaluate the full environmental modulation of the GCCN-ISP mechanism.

Manuscript revision (Line 129):

"As mentioned earlier, in this equation, there is no new water supplied by turbulent mixing, meaning that updrafts are not considered. In atmospheric clouds, the evolution of cloud droplet size is governed by two key factors: (1) the ambient water vapor pressure, which determines the vapor availability for condensation or deposition, and (2) the equilibrium saturation vapor pressure over the droplet surface, which is influenced by droplet size, curvature, and composition (via the solute effect). Updrafts affect cloud microphysics by cooling air parcels and enhancing supersaturation, thereby suppressing the WBF process and delaying ice growth (G.Abade et al., 2024; M.Guti'errez et al., 2023; M. Pinsky and A. Korolev., 2021). In contrast, the solute effect reduces the surface saturation vapor pressure, directly influencing whether supercooled droplets can persist. While both mechanisms can prolong the lifetime of mixed-phase clouds, this study intentionally excludes vertical motion to isolate the microphysical contribution of the solute effect to droplet–ice vapor competition."

4. Broader climatic implications not explored

Comment: While the mechanism's microphysical basis is plausible, its broader climatic impacts (e.g., on cloud radiative forcing or Arctic amplification) remain unquantified.

Response: We agree that linking the GCCN-ISP mechanism to cloud radiative effects is an important next step. Our current study focuses on the microphysical conditions that can extend droplet lifetimes, which could potentially impact cloud longevity and optical thickness. Although this work does not explicitly quantify radiative forcing, we now note in the revised

manuscript that the suppression of the WBF process by GCCN may increase cloud optical depth and extend lifetime, thereby influencing cloud feedbacks and Arctic amplification. Future research will aim to incorporate this mechanism into cloud and climate models.

Manuscript revision (Line 400):
"Finally, we emphasize the broader climatic relevance of the GCCN-ISP mechanism. The persistence of supercooled liquid water affects the radiative properties and lifetime of mixed-phase clouds. Suppressing the WBF process via solute effects could extend cloud longevity and enhance optical depth, potentially altering cloud radiative forcing and Arctic amplification. Although the present study does not explicitly quantify such impacts, we hope to explore these implications in future work by coupling the GCCN-ISP framework with climate models."

5. Competition/synergy with other ice pathways (e.g., SIP, coalescence)

Comment: The study does not explore how GCCN-ISP competes or synergizes with other ice nucleation pathways (e.g., secondary ice production) or collision-coalescence.

Response: Thank you for this insightful comment. We agree that interactions between GCCN-ISP and other ice-related processes such as secondary ice production (SIP) and collision–coalescence need discussion.

SIP mechanisms—particularly rime splintering—are known to be most active in the –3°C to –8°C temperature range (Korolev and Leisner, 2020). However, the cloud layer observed during the AFLUX campaign maintained top temperatures around –14.5°C, making SIP highly unlikely in this case. Therefore, we did not include SIP in our modeling framework.

As for collision–coalescence, this process predominantly governs droplet growth in warm clouds above 0°C and is generally ineffective in mixed-phase cloud regimes where ice processes dominate and droplet mobility is limited. Given that the cloud in our study was well below freezing and contained both droplets and ice, we consider the omission of warm-cloud collision–coalescence appropriate for the thermodynamic focus of this work.

Manuscript revision (Line 389):
"Secondary ice production (SIP) mechanisms, such as the Hallett–Mossop process, are generally restricted to a narrow temperature range of –3°C to –8°C (Korolev and Leisner, 2020). The cloud temperature analyzed in this study is around –14.5°C, making SIP unlikely. With this consideration, in the current modeling framework, the SIP process is excluded. Extending this work with more comprehensive models that incorporate ice particle shape, ventilation effects, aggregation, and turbulence will be important for assessing the robustness of this mechanism under realistic conditions."

6. Need for laboratory validation

Comment: Controlled laboratory experiments (e.g., cloud chamber studies) are critical to isolate GCCN-ISP under varying temperature, humidity, and CCN conditions.

Response: We strongly agree. Laboratory studies with controlled thermodynamic conditions, variable CCN size/composition, and co-located droplet/ice measurements would provide definitive tests of our theoretical framework. We have added a discussion on how cloud chamber platforms such as AIDA (Aerosol Interaction and Dynamics in the Atmosphere (Lamb et al., 2023)) and the Pi Chamber (Wang et al., 2024) can enable such studies. We particularly encourage future laboratory efforts to target Arctic-relevant conditions (T < –10°C) where solute effects may dominate and SIP is suppressed.

Manuscript revision (Line 394):
"Cloud chamber facilities such as AIDA (Aerosol Interaction and Dynamics in the Atmosphere (Lamb et al., 2023)) and the Pi Chamber (Wang et al., 2024) offer controlled environments to systematically vary CCN size, composition, and ambient conditions, allowing detailed observation of droplet–ice competition. While such platforms have been extensively used for

studying ice nucleation and SIP, targeted experiments focusing on solute-induced suppression of glaciation remain scarce. We encourage future laboratory activities that replicate conditions relevant to the Arctic to explore solute-driven persistence in supercooled droplets."

References:
Sorooshian, A., Siu, L. W., Butler, K., Brunke, M. A., Cairns, B., Chellappan, S., Chen, J., Choi, Y., Crosbie, E. C., Cutler, L., DiGangi, J. P., Diskin, G. S., Ferrare, R. A., Hair, J. W., Hostetler, C. A., Kirschler, S., Kleb, M. M., Li, X.-Y., Liu, H., McComiskey, A., Namdari, S., Painemal, D., Schlosser, J. S., Shingler, T., Shook, M. A., Silva, S., Sinclair, K., Jr., W. L. S., Soloff, C., Stamnes, S., Tang, S., Thornhill, K. L., Tornow, F., Tselioudis, G., Diedenhoven, B. V., Voigt, C., Vömel, H., Wang, H., Winstead, E. L., Xu, Y., Zeng, X., Zhang, B., Ziemba, L., and Zuidema, P.: The NASA ACTIVATE Mission, Bulletin of the American Meteorological Society, pp. BAMS–D–24–0136.1, https://doi.org/10.1175/BAMS-D-24-0136.1, 2025.

Behrenfeld, M. J., Moore, R. H., Hostetler, C. A., Graff, J., Gaube, P., Russell, L. M., Chen, G., Doney, S. C., Giovannoni, S., Liu, H., Proctor, C., Bolaños, L. M., Baetge, N., Davie-Martin, C., Westberry, T. K., Bates, T. S., Bell, T. G., Bidle, K. D., Boss, E. S., Brooks, S. D., Cairns, B., Carlson, C., Halsey, K., Harvey, E. L., Hu, C., Karp-Boss, L., Kleb, M., Menden-Deuer, S., Morison, F., Quinn, P. K., Scarino, A. J., Anderson, B., Chowdhary, J., Crosbie, E., Ferrare, R., Hair, J. W., Hu, Y., Janz, S., Redemann, J., Saltzman, E., Shook, M., Siegel, D. A., Wisthaler, A., Martin, M. Y., and Ziemba, L.: The North Atlantic Aerosol and Marine Ecosystem Study (NAAMES): Science Motive and Mission Overview, Frontiers in Marine Science, 6, https://doi.org/10.3389/fmars.2019.00122, journal Article, 2019.

Korolev, A. and Leisner, T.: Review of experimental studies of secondary ice production, Atmospheric Chemistry and Physics, 20, 11 767–11 797, https://doi.org/10.5194/acp-20-11767-2020, 2020.

Lamb, K. D., Harrington, J. Y., Clouser, B. W., Moyer, E. J., Sarkozy, L., Ebert, V., Möhler, O., and Saathoff, H.: Re-evaluating cloud chamber constraints on depositional ice growth in cirrus clouds – Part 1: Model description and sensitivity tests, Atmospheric Chemistry and Physics, 23, 6043–6064, https://doi.org/10.5194/acp-23-6043-2023, 2023.

Wang, A., Krueger, S., Chen, S., Ovchinnikov, M., Cantrell, W., and Shaw, R. A.: Glaciation of mixed-phase clouds: insights from bulk model and bin-microphysics large-eddy simulation informed by laboratory experiment, Atmospheric Chemistry and Physics, 24, 10 245–10 260, https://doi.org/10.5194/acp-24-10245-2024, 2024.

---

## Author Response (AR2)

**Response to Reviewer and Editor Comments**

We sincerely thank the editor and the reviewers for their constructive suggestions. We have carefully addressed all comments. Our point-by-point responses are detailed below, with changes implemented in the revised manuscript highlighted accordingly.

**Referee #2**

**Comment:**

My concern has to do with the applicability of the process to natural clouds with updrafts. I feel that the authors should add a comment such as that below to the Conclusions section, first paragraph: "While both mechanisms can prolong the lifetime of mixed-phase clouds, this study intentionally excludes vertical motion to isolate the microphysical contribution of the solute effect to droplet—ice vapor competition. However, the process needs to be studied both in models and natural clouds, with updrafts."

Response: We thank the reviewer for this important suggestion. We have added the suggested sentence to the first paragraph of the conclusions section as follow:

"While both mechanisms can change the lifetime of mixed-phase clouds, this study intentionally excludes vertical motion to isolate the microphysical contribution of the solute effect to droplet—ice vapor competition. However, the process needs to be studied both in models and natural clouds, with updrafts."

**Editor**

• Comment (Title and Lines 13, 121, 142, 154, 197, 213, 339, 349, 352, 371):

For consistency, "mixed phase" should be "mixed-phase"

Response: All instances of "mixed phase" have been corrected to "mixed-phase" for consistency.

• Comment (Line 50):

Replace "Giant Cloud Condensation Nuclei enhanced Ice Sublimation" by "Giant Cloud Condensation Nuclei-Enhanced Ice Sublimation Process"

Response: Corrected as suggested.

• Comment (Line 83):

Should "droplets cases" be "CCN cases"?

Response: Yes, this was a misstatement. It has been corrected to "CCN cases."

• Comment (Line 154):

Replace "small particle aerosols" by "small aerosol particles"

Response: Corrected as suggested.

• Comment (Lines 154-155):

Replace "condensation nuclei" by "CCN"

Response: Corrected as suggested.

• Comment (Line 164):

Replace "Bergeron process" by "WBF process"

Response: Corrected as suggested.

• Comment (Line 202):

Define AFLUX

Response: We have now added a definition for AFLUX: "...AFLUX (Airborne measurements of radiative and turbulent FLUXes of energy)..."

• Comment (Line 208):

Define "DLR"

Response: We have now added a definition for DLR "German Aerospace Center".

• Comment (Lines 220, 223 and 224):

Use consistent term (HYSPLIT)

Response: All mentions have been standardized to "HYSPLIT".

• Comment (Line 285):

Should "dry cloud droplet condensation nucleus diameter" be "dry cloud droplet CCN diameter"?

Response: Yes, and corrected as suggested.

• Comment (Lines 300, 316):

Replace "giant CCNs" by "GCCNs"

Response: All instances of "giant CCNs" have been changed to "GCCNs."

• Comment (Line 329):

The following statement is not completely clear: "The key parameter determines in which region is how much aerosol is actually inside the droplet"

Response: Revised to: "The key parameter determining which cloud droplets participate in the WBF process or the GCCN-ISP process is the size of the CCN within each droplet...".

• Comment (Line 334):

Define "INP"

Response: We have added the definition at first use "...INP (Ice Nucleating Particles)...".

• Comment (Line 367):

Replace "cloud condensation nuclei" by "CCN"

Response: Corrected as suggested.